# Effect of Electrode Induction Melting Gas Atomization on Powder Quality: Satellite Formation Mechanism and Pressure

**DOI:** 10.3390/ma16062499

**Published:** 2023-03-21

**Authors:** Jialun Wu, Min Xia, Junfeng Wang, Bo Zhao, Changchun Ge

**Affiliations:** Institute of Powder Metallurgy and Advanced Ceramics, University of Science & Technology Beijing, Beijing 100083, China

**Keywords:** EIGA, computational fluid dynamics, satellite powder, gas field

## Abstract

Electrode induction melting gas atomization (EIGA) is a wildly applied method for preparing ultra-clean and spherical metal powders, which is a completely crucible-free melting and atomization process. Based on several experiments, we found that although the sphericity of metal powders prepared by EIGA was higher than that of other atomization methods, there were still some satellite powders. To understand the formation mechanism of the satellite, a computational fluid dynamics (CFD) approach FLUENT and a discrete particle model (DPM) were developed to simulate the gas atomization process, and several EIGA experiments with different argon pressures (2.5–4.0 MPa) were designed. A numerical simulation of the gas-flow field verified the formation trajectory of satellites, and the Hall flow rate of the powder produced under different pressures was 13.3, 13.8, 15.6, and 16.8, which were consistent with the prediction of the numerical simulation. This study provides theoretical support for understanding the satellite formation mechanism and improving powder sphericity in the EIGA process.

## 1. Introduction

There is an increased demand for high-quality metallic powders used for additive manufacturing, such as in the aerospace, automotive, and medical fields and other applications [1]; this results in particular requirements for the production of spherical metal powders with narrow size distribution [2]. Various methods can prepare powder particles, including mechanical crushing, ball milling, gas atomization, electrolysis, and oxidation [3]. Metallic powders prepared by gas atomization have better sphericity and low oxygen content, mainly when the atomization is conducted under an inert gas atmosphere [4]. Due to the better kinetic energy transfer ability from gas to metal melt, argon atomization (AA) has been widely applied in recent years [5]. However, ceramic materials (crucibles and delivery tubes) in the AA equipment will easily react with alloys and bring certain amounts of non-metallic inclusions into the powders, which may seriously affect the performance of the produced components [6]. In the electrode induction melting gas atomization (EIGA) method, the melt flow is not in contact with refractories, which is often used to prepare ultra-clean metal powders [7,8].

The quality of gas-atomized powders is associated with two significant problems. The first is fine powder yield and particle size distribution, and the second is surface morphology and sphericity. In recent years, much research on EIGA has begun to appear. Geling [9], Cui [10], and Liu [11] studied EIGA parameters such as pressure and melting power but mainly focused on fine powder yield. Although powders prepared by EIGA have better sphericity than those prepared by other gas atomization processes such as VIGA (vacuum induction melting inert gas atomization) [12,13], there are still some satellites [14]. The satellites seriously reduce powder bulk density and fluidity and are prone to poor fusion in the forming process [15]. Some scholars focused on the problem of satellites in the gas atomization process. Özbilen [16] explored the satellite formation mechanism and observed the difference in satellite content under different materials and pressures, Achelis [17] used a gas recirculation system to reduce satellites, and Liu [18], Chen [19], and Xiao [2] studied powder morphology under different pressures. However, nearly no one mentioned the detailed satellite formation mechanism under different pressures. Therefore, it is necessary to understand the relationship between satellite and pressure in the EIGA process.

Since gas atomization is a complex physical process, where a high-speed gas quickly breaks a high-temperature fluid into tiny droplets, conventional experimental methods are inadequate to characterize the satellite formation process. Therefore, a computational fluid dynamics (CFD) approach was developed to simulate the gas atomization process. Zeoli [20] conducted numerical simulations to investigate the application and performance of close-coupled nozzles and studied the mechanism of droplet breakup in primary atomization. Bojarevics [1] investigated the complex interaction of the electromagnetic and thermal fields on the fluid flow with a free surface in the EIGA process by COMSOL software. Xia [21] designed a simulation of the entire gas atomization process and proved that the numerical simulation results agreed well with the experimental measurements. Djambazov [22] and Wang [23] proposed the reasons for forming satellites and hollow powders by CFD techniques. According to these studies, a visualization of the satellite formation in the atomization process can be achieved by numerical simulation.

This paper aims to understand the formation mechanism of satellites and explore the satellite content and fluidity of powder produced under different pressures. The experiments are performed on a self-developed EIGA device, and a simulation of the gas flow field and particle trajectory in the whole atomization chamber is performed using the commercial CFD software FLUENT. This work provides theoretical support for understanding the satellite formation mechanism under different pressures, thus, improving powder quality in EIGA.

## 2. Experimental and Simulation Methods

### 2.1. Experiments

In this work, experiments were performed on a self-developed EIGA device, Ni-based superalloy was selected as the metal in the experiments, and compressed argon was used as the atomization gas (Figure 1).

The diameter of the used Ni-based superalloy rod in Figure 1b was 55 mm, and the composition of the alloy is listed in Table 1.

Four different argon pressures were designed in the experiments to observe the satellite formation and content, which were 2.5, 3.0, 3.5, and 4.0 MPa. We controlled the continuous metal melt metal flow, as shown in Figure 1c, using specific process parameters, such as induction coil size and power supply, which could obtain fine powder yields [24], and all atomization parameters were set as 5 × 10^−3^ Pa for equipment vacuum, 48 ± 1 kW for smelting power, and 1970 ± 5 K for heating temperature.

The powder diameter distribution was analyzed using laser diffraction (BT-9300S), the flowability was measured using a powder tester (BT-1000) and metallographic microscope (RX50M), and the surface topography was observed using a scanning electron microscope (Regulus8100).

### 2.2. Numerical Simulation Parameters

The atomization chamber of the original EIGA equipment is an axisymmetric cylinder with a height of 2.6 m and a radius of 1 m, and, under the assumption that accuracy will not be affected, the model was simplified as a 2D model based on the symmetry of rotation to decrease the simulation workload. A mesh model (2.6 m × 1 m) was developed by AUTOCAD and GAMBIT 2.4.6 software to visualize the entire gas flow field better as shown in Figure 2, and it should be mentioned that the simulation was based on actual EIGA equipment drawings.

The initial mesh density was varied to increase calculation accuracy, we designed three different grids to verify the independence of the mesh, and the mesh numbers were 64,385, 166,772, and 47,950. Additionally, we chose three points (0, 0; 80, 10; 125, 15) to observe the changes in the velocity in different grids, which are shown in Figure 3.

The velocity of three points was basically stable when the number of model grids had been increased from 64,385 to 166,772, so the number of grids, 166,772, for gas atomization simulation was independent. In this condition, different density grids were divided at different zones to keep a balance between the calculation efficiency and accuracy. The density around the nozzle area was high (from 0.1 mm to 0.4 mm) and lowered in the other areas (from 0.4 mm to 4 mm), and the total number of grids was 166,772 under this design condition. Then, the boundary conditions, such as axis, wall, pressure inlet (argon), and pressure outlet, were defined, and the 2D-mesh file was exported and imported into FLUENT 2020 R2 software, where the remaining parameters were set. A simulation of the stable single-phase air flow field (argon) was conducted first, and the physical properties of argon were taken from the FLUENT database and are listed in Table 2. The k-ε turbulence model was employed, and the gas pressures were set as 2.5, 3.0, 3.5, and 4.0 MPa. Furthermore, the second-order upwind scheme was selected from the possible schemes, the time step size was set as 6 × 10^−7^, and the number of time steps was 10^6^.

Generally, the whole gas atomization goes through two interrelated stages: primary atomization and secondary atomization, and the satellite is mainly formed in the secondary atomization stage [2]. The discrete particle model (DPM) and Taylor analogy breakup (TAB) model under 3 MPa were applied in FLUENT software to observe the particle’s trajectory in the atomization chamber and explore the satellite formation mechanism. The diameter distribution of particles was set as from 1 mm to 0.25 mm, the position was from (2564, 0) to (2584, 13), and the physical properties of the Ni-based superalloy are listed in Table 3 [21], and the DPM model was calculated based on the completed single-phase argon field model.

### 2.3. Theoretical Model

During various solidification stages, the satellites are produced by collisions or impingement of fine solidified powder particles into the coarser molten or semi-molten particles. According to Yang’s research, the particles are assumed as a rigid sphere at the nozzle exit, and the cooling rate of particles with different diameters is deduced as follows [26]:(1)$$\left|\frac{dT_{d}}{dt}\right|=\frac{6}{\rho c_{P}}\left(T_{d}-T_{f}\right)\left(\frac{2k_{g}}{d^{2}}+0.6{\sqrt{R_{e}}}^{3}\sqrt{P_{r}}\right)$$
where $T_{d}$ is the temperature of particles; *t* is the cooling time; ρ, $c_{P}$, d, $R_{e}$, and $P_{r}$ are the density, specific heat, diameter, and Reynolds number and Prandtl number of particles, respectively; $T_{f}$ and $k_{g}$ are the temperature and thermal conductivity of argon.

$R_{e}$ can be defined as:(2)$$R_{e}=\frac{\rho vD}{\mu }$$
where ρ, v, and μ are the density, velocity, and viscosity of the particles; D is the diameter of the continuous metal flow, which is 4 mm [27].

$P_{r}$ can be defined as:(3)$$p_{r}=\frac{c_{p}\mu }{\lambda }$$
where $c_{p}$, μ, and λ are the specific heat, viscosity, and thermal conductivity of the particles.

According to the parameters in Table 2 and Table 3, and Equations (1)–(3) can be calculated as follows:(4)$$\left|\frac{dT_{d}}{dt}\right|=\left(\frac{12k_{g}}{\rho c_{P}d^{2}}+\frac{3.6}{\mu }\sqrt{\frac{\rho v^{3}D^{3}}{\lambda c_{p}}}\right)\left(T_{d}-T_{f}\right)$$

In the process of simulation, the standard k-ε turbulence model was employed in FLUENT, which is suitable for complex flows, such as a high Reynolds number, mixed flow, and swirl [28]. The mass equation, continuity equation, momentum equation, and energy equation are given in the following [29,30,31]:

Mass equation
(5)$$\frac{\partial \rho }{\partial t}+\nabla \cdot \left(\rho \vec{u}\right)=0$$

Continuity equation
(6)$$\frac{\partial \rho }{\partial t}+\frac{\partial }{\partial x}\left(\rho u_{x}\right)+\frac{\partial }{\partial r}\left(\rho u_{r}\right)+\frac{\rho u_{r}}{r}=0$$

Momentum equation
(7)$$\frac{\partial }{\partial t}\left(\rho \vec{u}\right)+\nabla \cdot \left(\rho \vec{u}\vec{u}\right)=-\nabla P+\nabla \cdot \left[\overset{=}{\tau }\right]+\rho \vec{g}+\vec{F}$$

Energy equation
(8)$$\frac{\partial }{\partial t}\left(\rho E\right)+\nabla \cdot \left(\vec{u}\left(\rho E+P\right)\right)=-\nabla \cdot \left(\underset{j}{\sum }h_{j}J_{j}\right)+S_{h}$$

The Euler–Lagrangian discrete phase model was employed to observe the particle’s trajectory in secondary atomization, where the aerodynamic coefficients Weber number (*We*) was an essential dimensionless parameter. With the increase in *We*, a variety of typical secondary crushing patterns, including Taylor analogy breakup (TAB) models, Wave models, and Kelvin–Helmholtz (KH) models, appear [20].

*We* can be defined as:(9)$$We=\frac{\rho_{g}U^{2}d_{L}}{\sigma }$$

According to the parameter settings in DPM, *We* was calculated as ranging from 25 to 101, which determines that the TAB model should be selected [32].

## 3. Results and Discussion

### 3.1. Satellite Formation

In the EIGA experiment, obvious recirculation can be found from the observation window. Figure 4 and Figure 5 show the recirculation phenomenon and SEM photos of powder in the EIGA process with 3.0 MPa pressure (no. 2).

In Figure 4, the recirculation phenomenon is recorded during the EIGA process. Frame-by-frame display with camera showed the minor fragments in the equipment raised with the airflow and moved to the central area during the atomization process in Figure 4b–f, which indicates that there was an extensive range of recirculation zone in the atomization equipment, and this phenomenon might be the leading cause of satellites.

The morphology of the atomized powders and their representative particles are illustrated in Figure 5. It can be seen that most of the powders were nearly spherical with a few irregular-shaped powder particles, and some larger particles were decorated with satellites on the surface in Figure 5b–d.

Based on the calculation results of DPM, 6,890,000 particles were simulated by FLUENT, and the velocity, temperature, and diameter of the droplet were calculated to observe the particle’s trajectory in the atomization chamber and explore the satellite formation mechanism. The trajectories of atomized powders are shown in Figure 6, and the particles’ temperature, velocity, and diameter are shown in Figure 7 (t = 1.1873 s).

As shown in Figure 6, obvious particle recirculation was observed in the whole atomization chamber, which could mainly be classified into four steps. In the first step, the initial particles moved down and broke up with the high-speed argon until they came into contact with the bottom of the chamber (0.2769 s–0.3123 s). In the second step, the initial particles hit the bottom at a certain speed under argon and gravity and moved horizontally along the bottom after bouncing (0.3123 s–0.4034 s). In the third step, the initial particles moved upward along the chamber wall under the action of argon recirculation until the force provided by the argon was insufficient to resist the gravity of the particles themselves (0.4034 s–0.8993 s). In the furth step, the upward kinetic energy of the initial particles ran out, and the particles began to move towards the center, relying on the argon recirculation zone, then merged into the particle stream produced in the first step and collided with it to form satellite powders. Figure 6 demonstrates the presence of particle recirculation and the possibility of satellite formation.

As shown in Figure 6b,g, the initial particles contacted the subsequently atomized particles at the dotted line, the time from (a) to (b) was 0.0107 s, and, in this process, the particle temperature decreased from 1800 K to 1200 K. From Figure 7a, it can be observed that the particle velocity in this process was assumed as 150 m/s according to Figure 7b, and the particle diameter calculated from Figure 7c was 50.6 μm, and Equation (4) can be calculated as follows:(10)$$\left|\frac{dT_{d}}{dt}\right|=\left(\frac{12\times 0.0158}{7056\times 720\times d^{2}}+\frac{3.6}{0.05}\sqrt{\frac{7056\times 150^{3}\times 0.004^{3}}{29.6\times 720}}\right)\times \left(1970-300\right)=\frac{6.23\times 10^{-5}}{d^{2}}+3.22\times 10^{4}$$

According to Equation (10), the cooling rate was calculated as 5.65 × 10^4^ K/s, and the particle temperature dropped by 604.4 K in the process from Figure 6a to Figure 6b, which is consistent with the particle temperature variable result in Figure 7a.

As seen in the simulation result shown in Figure 7b, the velocity of initial particles after recirculating was negligible, the relative velocity between initial particles and subsequently atomized particles was assumed as 100 m/s, and the low-temperature particles collided with the high-temperature particles to form satellites.

### 3.2. Argon Recirculation Velocity Field in Different Pressure

The particle recirculation phenomenon visually indicated the formation process of satellite powder, and it can be speculated that the strength of the recirculation zone was a critical point in the satellite formation process. A single-phase model was adopted to investigate the recirculation zone characteristics at different atomization pressures, as shown in Figure 8 and Figure 9.

Figure 8 shows the velocity field at four different atomization pressures; the flow field was a symmetrical structure. Figure 8a–c and the left side of (d) are the steamtraces velocity fields, and the streamtraces demonstrated a large recirculation zone below the device, which caused the atomized powders to form satellites. The intersection of the dotted line is the center of the recirculation zone, and the velocity near the wall increased with argon pressure. It was found that the trend of the argon field on the dotted line was the same, but the recirculation velocity at high pressure (4.0 MPa) was significantly higher than that at low pressure (2.5 MPa). The right part of Figure 8d is the argon velocity contour, and a supersonic speed can be seen in the gas outlet, which provided the atomized powder with a high initial velocity and increased relative velocity to recirculated particles and made it easier to combine into satellites.

Figure 9 shows the recirculation zone characteristics at different atomization pressures, and the accurate values were provided by simulation results. The centric position of the recirculation zone is shown in Figure 9a, the X position increased (from 334 to 412 mm), and the Y position decreased (from 729 to 708 mm) with the argon pressure increase, which indicates that the centric position gradually approached the primary atomization area and that the probability of satellite formation might increase as the pressure increases. The recirculation velocity and max velocity are presented in Figure 9b, the recirculation velocities were 11.7, 12.4, 13.7, and 15 m/s, and the increased ratios were 6.0%, 10.5%, and 9.5% from 2.5 to 4.0 MPa. It can be found that the change in recirculation velocity will be stable at about 10% with the pressure increase; the max velocities (gas outlet velocity) were 487, 504, 509, and 513 m/s; the increased ratios were 3.5%, 1.0%, and 0.8% from 2.5 to 4.0 MPa; and the speed was stable as the pressure increased.

According to the comparison between Figure 6 and Figure 8b, the particle trajectory coincided nicely with argon steamtraces, which indicates that the argon recirculation was the critical point for particle recirculation and satellite formation. The characteristics in Figure 9 also show that as the pressure increased, the recirculation velocity affecting the formation of the satellite increased much more than the max velocity, which would inevitably lead to an increase in the number and velocity of recirculated particles.

Combined with the intersection lines of the initial particles and the subsequently atomized particles in Figure 6 and Figure 7, it can be speculated that the X position of the intersection line must rise with the pressure, velocity, and temperature of the subsequently atomized particles that collide with the recirculated particles are further increased, which will increase the degree and probability of bonding, and it can be inferred that the satellite content will increase with the pressure.

### 3.3. Atomized Powder Detection

According to the atomization parameters, EIGA atomization experiments were designed with different pressures (2.5, 3.0, 3.5, and 4.0 MPa), further verifying the relationship between the satellites and atomization pressure.

Figure 10 shows the SEM images of atomized powders and the distribution curves. As shown in Figure 10a–d, the atomized powders sizes were increased with the atomization pressure, the size produced for the pressure of 4.0 MPa was smaller than that for the 2.5 MPa, and there were a small number of large particle powders for 2.5 MPa. It was visually observed that the powder sphericity was different for the four atomization pressures, the atomized powder was nearly spherical for 2.5 MPa and 3.0 MPa, but the sphericity of the powders prepared at 3.5 MPa began to decrease, and the content of the visible satellite increased with the atomization pressure. According to the results of the laser particle analyzer shown in Figure 10e, it was visually observed that the powder size decreased with the increase in pressure, and the D_50_ values were 78.58, 66.27, 58.07, and 43.19 μm for the pressures of 2.5, 3.0, 3.5, and 4.0 MPa, respectively.

In order to further verify the influence of atomization pressure on the sphericity of the powder, the properties of the powder were further tested by the powder tester (BT-1000), and the apparent density, tapped density, flow index, and Hall flow rate are summarized in Table 4. The Carr compressibility and Hausner’s ratio are two indicators of powder flow index; the smaller the two are, the better the fluidity of powder is, and they are deduced as follows [33]:(11)$$\mathrm{Carr}\;\mathrm{compressibility}=\frac{\mathrm{apparent}\;\mathrm{density}\;-\;\mathrm{tapped}\;\mathrm{density}}{\mathrm{tapped}\;\mathrm{density}}\times 100\%$$
(12)$$\mathrm{Hausner}’s\;\mathrm{Ratio}=\frac{\;\mathrm{tapped}\;\mathrm{density}}{\mathrm{apparent}\;\mathrm{density}}$$

Table 4 summarizes the relevant parameters of powder sphericity for different pressures. It was found that the apparent density increased with the atomization pressure when it was less than 3.5 MPa, and decreased when the pressure increased to 4.0 MPa, which indicates that the particle size decreased with the increased pressure, but the content of the satellite increased when the pressure was too high (4.0 MPa), which inhibited the trend of increasing apparent density.

As illustrated in Table 4, the flow index gradually weakened with the increase in pressure, and the powder fluidity evaluation is unsuitable for the pressure of 4 MPa, in which the Carr compressibility exceeded 16%, and the Hausner’s Ratio exceeded 1.19 (ASTM D6393). Moreover, the Hall flow rate also indicates that the powder fluidity decreased with the pressure increase.

In summary, the testing of powder properties verified that powder fluidity decreased when pressure increased, which further indicates that the content of satellite powder increased, and the results proved the correctness of the numerical simulation prediction in Section 3.2. It can be concluded that the choice of pressure was not the higher the better in the gas atomization process, and the adverse effects caused by the increase in satellite content and the decrease in powder fluidity under high pressure should be fully considered.

## 4. Conclusions

This work performed CFD simulations to investigate the formation mechanism of the satellite and the pressure influence on powder fluidity in the EIGA process, and the obtained results agreed well with the experimental data. More specifically, the k-ε turbulence model, the discrete particle model (DPM), and Taylor analogy breakup (TAB) model were adapted to simulate the single-phase argon flow field and particle trajectory, which explored the satellite formation mechanism. Moreover, several EIGA experiments with different argon pressures (2.5–4.0 MPa) were designed to observe the powder morphology and properties. This study provides theoretical support for understanding the satellite formation mechanism and improving powder sphericity in the EIGA process:A numerical simulation of particle trajectory verified the formation of satellites, and the simulation results were consistent with the experimental photos. Moreover, the decreasing trend of particle temperature in the numerical simulation was verified by the particle cooling equation, and the accuracy of the simulation work was verified.A numerical simulation of the single-phase argon flow field verified the difference in the recirculation zone characteristics at different atomization pressures. The centric position gradually approached the primary atomization area (X from 334 to 412 mm and Y from 729 to 708 mm), and the recirculation velocities increased from 11.7 to 15 m/s, which provides a theoretical basis for the increase in satellite content in high-pressure atomization.The detection of atomized powder under different pressures proved the decrease in powder fluidity prepared under high pressure. The Carr compressibility and Hausner’s ratio increased from 11% to 18% and from 1.13 to 1.22, and the Hall flow rate increased from 13.3 to 16.8 s/(50 g), which indicates that the powder fluidity decreased with the pressure increase.Moreover, this work indicates that the selection of pressure should consider the adverse effects caused by the increase in satellite content and the decrease in powder quality under high pressure, and the appropriate atomization pressure should be selected comprehensively.

## Figures and Tables

**Figure 1 materials-16-02499-f001:**
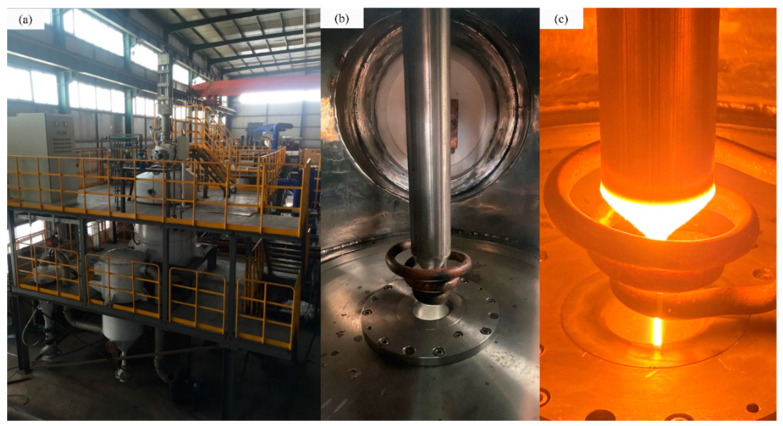
Self-developed EIGA device. (**a**) EIGA device, (**b**) melting chamber, and (**c**) continuous melting.

**Figure 2 materials-16-02499-f002:**
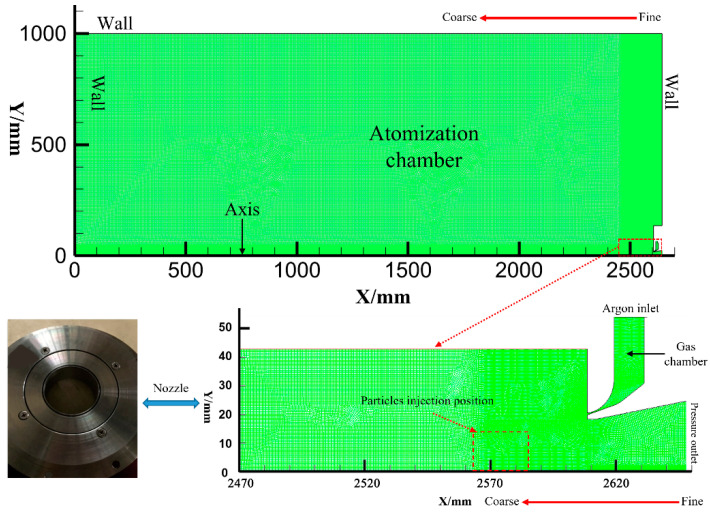
Mesh model of the EIGA device.

**Figure 3 materials-16-02499-f003:**
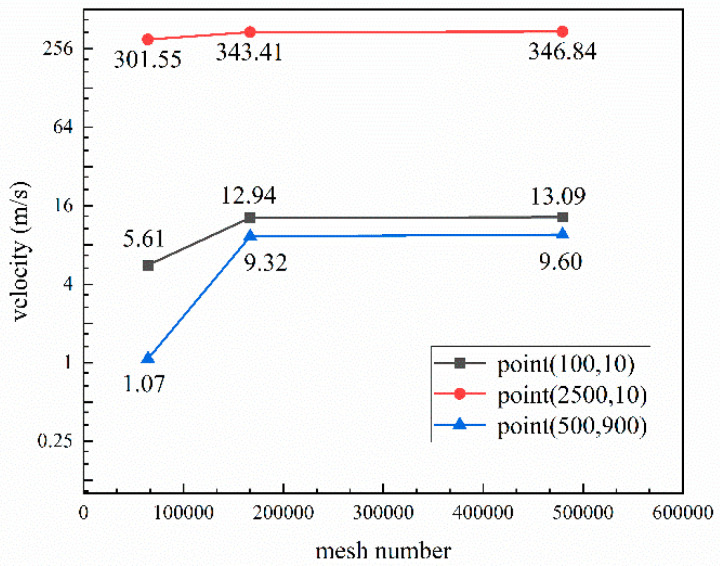
Gas velocity under different model grid numbers.

**Figure 4 materials-16-02499-f004:**
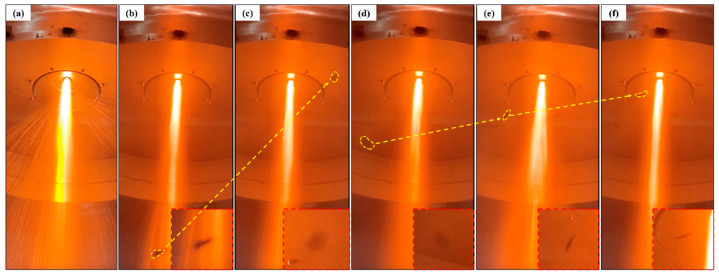
The recirculation phenomenon photos in EIGA equipment. (**a**) Atomization phenomenon (**b**,**c**) and (**d**–**f**) recirculation phenomenon in the EIGA device.

**Figure 5 materials-16-02499-f005:**
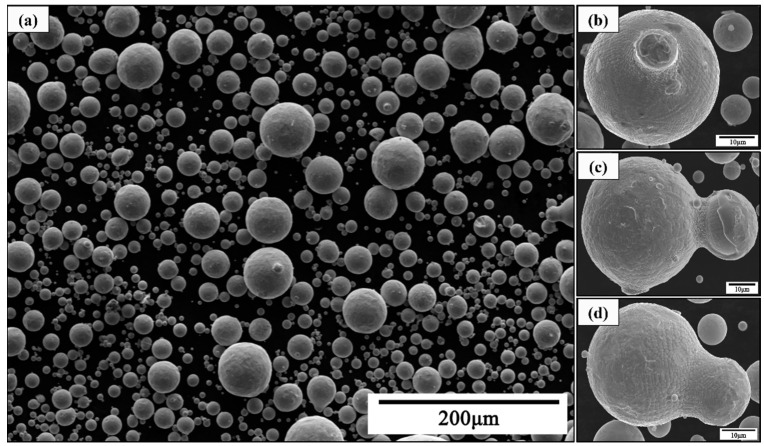
SEM images of powders under 3 MPa pressure. (**a**) Morphology of powders, (**b**–**d**) morphology of different satellites.

**Figure 6 materials-16-02499-f006:**
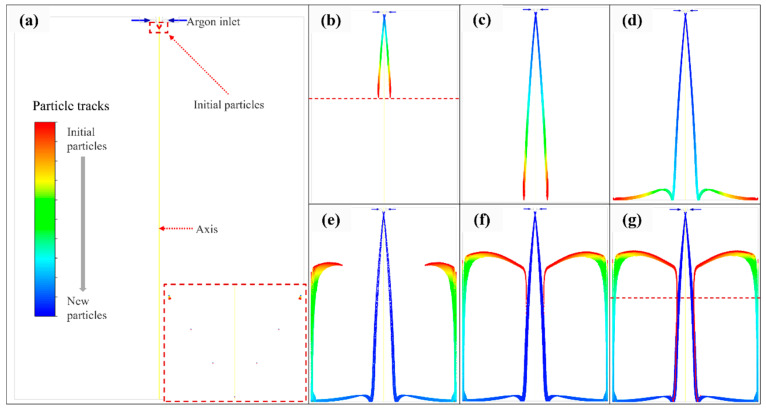
Particle trajectories. (**a**) 0.2769 s, (**b**) 0.2876 s, (**c**) 0.3123 s, (**d**) 0.4034 s (**e**) 0.8993 s, (**f**) 1.1713 s, and (**g**) 1.2273 s.

**Figure 7 materials-16-02499-f007:**
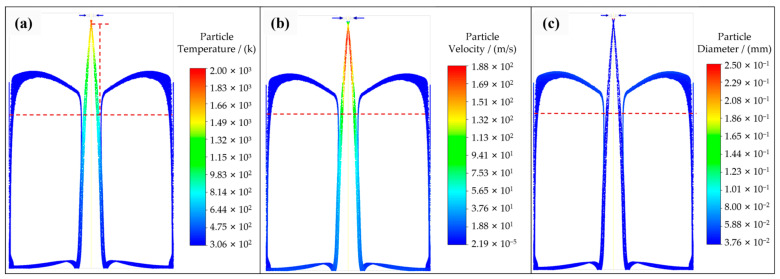
Particle variables. (**a**) Temperature, (**b**) velocity, and (**c**) diameter.

**Figure 8 materials-16-02499-f008:**
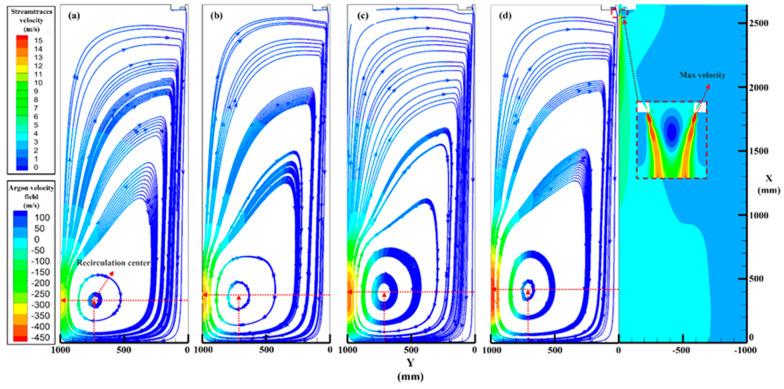
Argon velocity fields. (**a**) 2.5 MPa, (**b**) 3.0 MPa, (**c**) 3.5 MPa, and (**d**) 4.0 MPa.

**Figure 9 materials-16-02499-f009:**
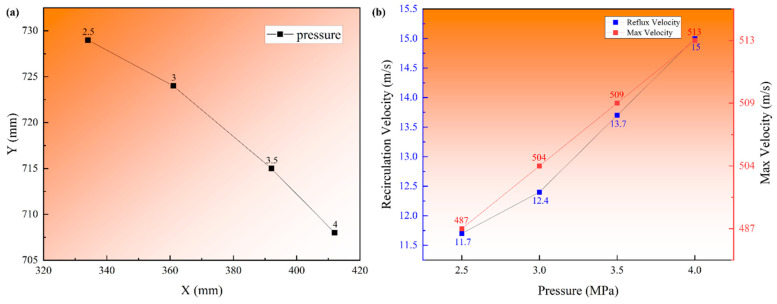
The recirculation zone characteristics at different atomization pressures. (**a**) Centric position, (**b**) recirculation velocity and max velocity.

**Figure 10 materials-16-02499-f010:**
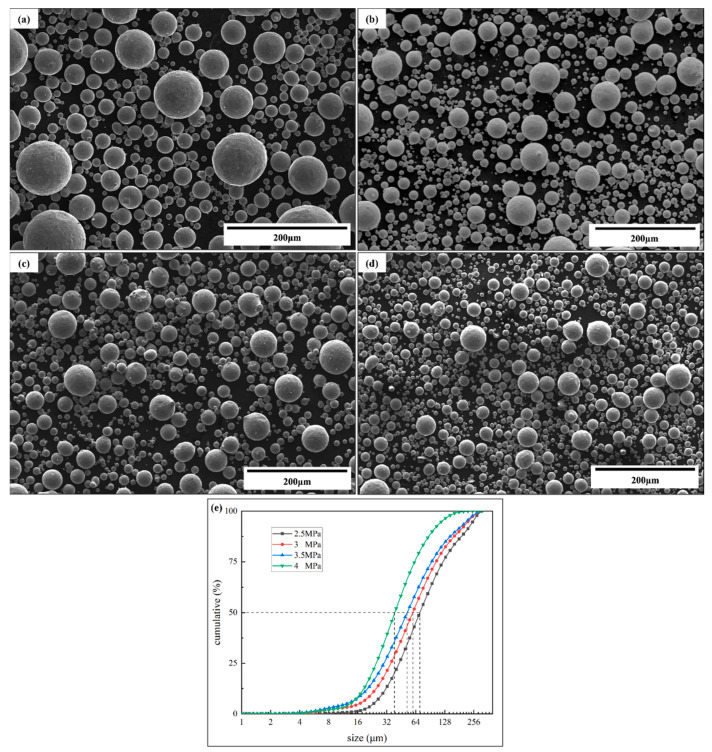
SEM images and distributions of atomized powders. (**a**) 2.5 MPa, (**b**) 3.0 MPa, (**c**) 3.5 MPa, (**d**) 4.0 MPa, and (**e**) particle size distribution.

**Table 1 materials-16-02499-t001:** Composition and content of Ni-based superalloy.

Component	Ni	Co	Cr	W	Ti	Al	Mo	Ta	Nb	C	B	Zr
Content/%	46.1	21.7	13.4	4.73	3.58	3.49	2.67	1.59	1.70	0.046	0.025	0.063

**Table 2 materials-16-02499-t002:** Physical properties of argon.

Parameter	Value	Unit
Density	1.63	kg·m^−3^
Cp (specific heat)	520	j·kg^−1^·k^−1^
Thermal conductivity	1.58 × 10^−2^	w·m^−1^·k^−1^
Viscosity	2.13 × 10^−5^	kg·m^−1^·s^−1^
Molecular weight	39.9	kg·kmol^−1^
Temperature	300	K

**Table 3 materials-16-02499-t003:** Physical properties of the Ni-based superalloy [25].

Parameter	Value	Unit
Specific heat	720	J·kg^−1^·K^−1^
Thermal conductivity	29.6	W·m^−1^·K^−1^
Viscosity	0.05	mPa·s
Surface tension	1.84	mN·m^−1^
Density	7056	kg·m^−3^
Melting point	1683	K
Solidification interval	1683–1823	K

**Table 4 materials-16-02499-t004:** Comparison of powder properties from different pressures.

No.	Argon Pressure/MPa	Powder Size/μm	Apparent Density/g·cm^−3^	Tapped Density/g·cm^−3^	Flow Index		Hall Flow Rate/s·(50 g)^−1^
					Carr Compressibility	Hausner’s Ratio	
1	2.5	D_50_ = 78.58	5.12	5.76	11%	1.13	13.3
2	3.0	D_50_ = 66.27	5.19	5.88	12%	1.13	13.8
3	3.5	D_50_ = 58.07	5.27	6.12	14%	1.16	15.6
4	4.0	D_50_ = 43.19	5.20	6.32	18%	1.22	16.8

## Data Availability

The datasets generated and analyzed during the current study are available from the corresponding author upon reasonable request.

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
