# Peer review of "Effect of Electrode Induction Melting Gas Atomization on Powder Quality: Satellite Formation Mechanism and Pressure"

_materials, 2023, doi:10.3390/ma16062499_

Round 1

Reviewer 1 Report

1.       Please mention the CFD solver used to simulate the work in the abstract.

2.       The GAMBIT is an ancient software to develop the meshed model. Please mention the valid license of the GAMBIT version.

3.       It’s mandatory to provide the Grid sensitivity analysis for the CFD work.

4.       There is no info regarding the DPM meshed model and boundary conditions.

5.       Please provide the work to validate the computational works.

6.       The velocity vector is crucial to explain particle movement.

7.       Please explain the reason for providing an SEM image as in Figure 9.

Reviewer 2 Report

The papers deals with the discussion of the numerical simulation of the powderization process by electrode-induction melting gas atomization. The paper is clearly written and the results are reliable. I believe that the paper may be published upon correcting the following issues:

Table 2: most data are the same for all the conditions and can be moved to a header.

Tables 2 and 5 and throughout the text: The same number of significant digits should be presented for the pressure

Table 3: Data sources should be mentioned.

Table 3: The numbers of significant digits of 5 are inappropriate for the selected type of measurements, it should be much less. If the values were considered as constants for the calculation, the maximum uncertainty of these values as reference data should be provided.

The number of significant digits should ve checked and corrected according to the overall uncertainty level of the presented approach (it seems that 3 significant digits is the appropriate precision level)

Figures 3 and 4, the captions should contain brief descriptions of individual images, now they are in the text only.

Some grammar and especially punctuation errors should be corrected.

Reviewer 3 Report

The authors studied the satellite formation mechanism in powder produced by electrode induction melting gas atomization. While the manuscript is generally well executed, there are several issues that should be addressed before further consideration for publication.

1. In the Introduction, additive manufacturing is listed together with other industries, such as aerospace. However, they are manufacturing techniques? Please clarify.

2. How does Figure 3 showed recirculation?

3. How are the simulations validated if there is no observation of the flow during the EIGA process? 

4. By measuring the flowability of the powder and particle distribution size, any discussion on the extent of the satellite formation? 

Round 2

Reviewer 3 Report

NIL